# Multiple Mycotoxin Contamination in Livestock Feed: Implications for Animal Health, Productivity, and Food Safety

**DOI:** 10.3390/toxins17080365

**Published:** 2025-07-25

**Authors:** Oluwakamisi F. Akinmoladun, Fabia N. Fon, Queenta Nji, Oluwaseun O. Adeniji, Emmanuel K. Tangni, Patrick B. Njobeh

**Affiliations:** 1Department of Biotechnology and Food Technology, Faculty of Science, University of Johannesburg, Johannesburg 2006, Gauteng, South Africaseunola2012@gmail.com (O.O.A.); pnjobeh@uj.ac.za (P.B.N.); 2Department of Agriculture, University of Zululand, KwaDlangezwa 3886, KwaZulu-Natal, South Africa; fonf@unizulu.ac.za; 3Sciensano, Chemical and Physical Health Risks Organic Contaminants, Toxins Unit, Leuvensesteenweg 17, 3080 Tervuren, Belgium; emmanuel.tangni@sciensano.be

**Keywords:** mycotoxins, fungi, mycotoxicosis, multi-occurrence, livestock

## Abstract

Mycotoxins are toxic secondary metabolites produced by various fungi that contaminate livestock feed, posing serious threats to animal health, productivity, and food safety. Although historical research has often examined individual mycotoxins in isolation, real-world conditions typically involve the simultaneous presence of multiple mycotoxins, resulting in additive or synergistic toxic effects that are often more severe than those observed with single toxin exposures. This review comprehensively synthesizes recent findings on multi-mycotoxin contamination in livestock feed, highlighting their physiological effects, mechanisms of action, and implications for regulatory frameworks. Multi-mycotoxin interactions exacerbate oxidative stress, immune suppression, impaired reproduction, and organ damage across species, leading to reduced growth performance, decreased milk and egg production, compromised carcass and wool quality, and increased mortality rates. A major concern is that current international regulatory standards mainly address individual mycotoxins, overlooking the compounded risks of co-occurrence. Global surveillance studies consistently reveal high prevalence rates of mycotoxin mixtures in feedstuffs, especially combinations involving DON, ZEN, AFB_1_, FB_1_, and OTA. Understanding these interactions and their underlying cellular mechanisms is critical for improving risk assessment models, formulating integrated mitigation strategies, and safeguarding both livestock productivity and human food security.

## 1. Introduction

Mycotoxins are naturally occurring toxic compounds in food and feed produced by various fungi under favourable environmental conditions, either in the field (pre-harvest) or during storage (post harvest) [1]. They represent a chemically diverse group of secondary metabolites, with some of the most significant including aflatoxins (AFs) from *Aspergillus flavus*; ochratoxin A (OTA) from *Aspergillus ochraceus* and *Penicillium verrucosum*; and several toxins such as deoxynivalenol (DON), zearalenone (ZEN), fumonisins (FUM), and trichothecenes produced by *Fusarium* species [2]. Recently, emerging mycotoxins such as beauvericin (BEA) and enniatins (ENNs) have also garnered attention due to their occasional detection in food and feed [3].

Mycotoxins are prevalent in cereals, nuts, and their by-products, often accumulating in compounded livestock feeds at higher concentrations [4]. The ingestion of contaminated feed serves as a key route for mycotoxins into livestock and their products (e.g., eggs, organs, meat, and milk), posing a significant public health risk. When ingested by animals above a certain concentration limit, the response effect is termed mycotoxicosis, eliciting varying degrees of symptoms from acute to chronic depending on the physiological status. Under field conditions, the symptoms of mycotoxin-induced toxicity in animals range from reduced productivity, immunosuppression, heightened disease susceptibility, and, in several cases, death [5].

To safeguard animal and human health, numerous countries have established regulatory thresholds for individual mycotoxins [6,7]. However, under field conditions, mycotoxins rarely occur in isolation. Co-contamination, driven by the co-infection of crops by multiple fungal species or environmental factors exacerbated by climate change and global feed trade, has become increasingly prevalent [8,9].

Historically, research on mycotoxins has focused on individual toxins. However, natural contamination often involves multiple mycotoxins due to co-infection by different fungi or co-contamination across commodities. This polytoxicity poses a heightened risk, as additive or synergistic interactions can intensify adverse health effects even when individual toxin levels remain below the regulatory limits [10]. Yet, the current regulatory frameworks and risk assessment models largely overlook the complexities of multi-mycotoxin interactions.

This review critically synthesizes the current knowledge on the prevalence, toxicological mechanisms, interaction effects, and animal health impacts of multi-mycotoxin contamination in livestock feed, while also highlighting current research gaps and opportunities.

## 2. Mycotoxins of Economic Importance: Regulatory Framework and Global (Co-)Occurrence

### 2.1. Key Mycotoxins in Animal Agriculture

Aflatoxins (AFs), primarily produced by *A. flavus* and *A. parasiticus*, frequently contaminate high-oil-content crops such as maize, peanuts, and cottonseed [11,12]. Common AFs include B_1_, B_2_, G_1_, G_2_, and the hydroxylated metabolite M_1_. These toxins are chemically stable, highly toxigenic, and classified as Group 1 carcinogens by the International Agency for Research on Cancer [13]. Among them, AFB_1_ is the most potent, estimated to be 68 times more toxic than arsenic and 10 times more toxic than potassium cyanide [14]. It primarily targets the liver, but its effects extend to reduced weight gain, diminished milk and egg production, impaired feed efficiency, and increased disease susceptibility [15].

Fumonisins (FUMs) are produced by *F. verticillioides* and *F. proliferatum*. Maize is the primary crop contaminated by fumonisins, and contamination usually occurs during the pre-harvest and post-harvest stages, especially under warm and dry conditions [16]. Common FUMs include FB_1_, FB_2_, and FB_3_, with FB_1_ being the most toxic [17]. FB_1_ exhibits hepatotoxic and nephrotoxic properties and is carcinogenic in animals [18]. Among livestock, pigs show the highest sensitivity, while poultry and ruminants exhibit relative resistance. In pigs, FB_1_ exposure can induce pleural effusion, pulmonary oedema, and pancreatic damage [19].

Zearalenone (ZEN), produced by *Fusarium graminearum* and *Fusarium culmorum,* commonly affects crops like maize, wheat, barley, oats, and sorghum [20]. ZEN is known for its strong estrogenic and anabolic activity, primarily targeting the reproductive organs. It disrupts gonadal and endocrine functions in both humans and animals, leading to genetic toxicity, hepatotoxicity, and immunotoxicity [21]. Cases of estrogen hyperactivity, nervous system overstimulation, abortion, stillbirth, and fetal abnormalities have been reported following the consumption of ZEN-contaminated feed [22]. Pigs are particularly sensitive, with symptoms including swelling of the vulva, nipples, vagina, and uterus, as well as vaginal or rectal prolapse in severe cases [23]. According to [24], a diet containing 1.5 mg/kg ZEA in pigs can induce estrogenic syndrome within 3–7 days.

Trichothecenes, produced by *Fusarium graminearum*, *Fusarium culmorum*, *Fusarium sporotrichioides*, and *Fusarium poae*, affect wheat, barley, oats, maize, and other cereal grains [25]. They are classified into four groups (A, B, C, and D) based on their structure and functional groups, with type A (e.g., HT-2 toxin, T-2) and type B (e.g., deoxynivalenol, DON) being the most prevalent. DON targets the gastrointestinal tract and immune systems, with pigs being particularly vulnerable, absorbing up to 55% of the toxin through the intestine [26]. Toxic effects can manifest at concentrations as low as 1–2 mg/kg, leading to reduced feed intake and nutrient utilization, gastrointestinal distress (vomiting, nausea, diarrhea, and anorexia), and immune dysfunction [27]. Additionally, DON can form a masked mycotoxin, deoxynivalenol-3-glucose (D-3-G), when it binds with glucose in grains [28]. The T-2 toxin has been linked to immunosuppression, feed refusal, gastroenteritis, intestinal hemorrhages, production losses, and mortality in calves and dairy cattle [29]. In poultry, T-2 toxin causes mouth and intestinal lesions, impairs immune function, reduces egg production, decreases feed intake, and leads to weight loss and feather abnormalities [30].

Ochratoxins (OTAs), mainly produced by *A. ochraceus*, *A. carbonarius* (tropical climates), and *P. verrucosum* (temperate climates), affect cereals (wheat, barley, oats), grapes, coffee beans, dried fruits, and spices. OTA is produced under improper storage conditions and can contaminate a variety of plant-based products [31]. OTA inhibits protein synthesis, disrupts cell-cycle progression, and induces DNA adduct formation [32]. OTAs also suppress ATP production and trigger the release of reactive oxygen and nitrogen species in the mitochondria [33]. Table 1 is a summary of mycotoxin target organs and their effects on different livestock species.

### 2.2. Regulatory Framework for Mycotoxin in Animal Feed

Regulatory frameworks for mycotoxin contamination in livestock feed are essential for safeguarding both animal health and food safety. Given the difficulty of completely eliminating mycotoxins from the food and feed chains, national and international agencies have established maximum allowable limits for individual mycotoxins in animal feed (Table 2). These limits are informed by scientific research on toxicity, exposure levels, analytical capabilities, and the economic impact of enforcement [42].

Historically, AFs were the first to be regulated, with limits introduced in the 1960s [43]. By 2003, over 100 countries had implemented regulatory thresholds for several mycotoxins [42]. In the United States, the Food and Drug Administration (FDA) established action levels for AFs and FBs, and advisory levels for DON [44,45,46]. Although no official limit exists for OTA in U.S. feed, the Joint FAO/WHO Expert Committee on Food Additives (JECFA) proposed a maximum of 5 µg/kg in cereals and by-products [47]. The European Union (EU), which enforces some of the strictest limits globally, sets a limit of 0.05 µg/kg for AFM_1_ in milk and 3–5 µg/kg for OTA in cereals [48,49].

Limits for ZEN in feed vary widely, from 50 to 1000 µg/kg, depending on the target species and feed type [48]. To support regulatory compliance and mitigate risk, the Codex Committee on Food Additives and Contaminants (CCFAC) recommends Good Agricultural Practices (GAP), Good Manufacturing Practices (GMP), and Hazard Analysis and Critical Control Points (HACCP) protocols [50].

Despite these efforts, a significant shortcoming of the current regulatory systems is their focus on individual mycotoxins, even though multi-mycotoxin co-occurrence is widespread. The existing standards do not consider the additive or synergistic effects that can arise when animals are simultaneously exposed to multiple mycotoxins, even at levels below the established individual limits. This regulatory gap is increasingly problematic, as growing evidence shows that co-contamination can amplify the adverse effects on animal health, productivity, and product quality. This has led to calls for a more holistic risk assessment approach that accounts for cumulative exposure and interaction effects. The high frequency of multi-mycotoxin contamination in livestock feed underscores the need for such an approach.

### 2.3. Prevalence of Multi-Mycotoxin Co-Occurrence in Livestock Feed

#### 2.3.1. Factors Influencing the Spread of Mycotoxin in Feed

A complex interplay of factors, including climate change, agricultural and storage practices, and the composition of raw feed materials, influences the contamination of agricultural commodities with mycotoxins.

Climate change plays a pivotal role by altering the environment’s abiotic and biotic conditions. These shifts create favourable conditions for fungal growth and lead to the emergence of previously unreported mycotoxins in new geographic regions [51]. For example, prolonged droughts caused by climate change can stress plants, weakening their natural defences and making them more susceptible to pest infestations, fungal infections, and, ultimately, mycotoxin contamination [52]. Fluctuating weather patterns leading to variation in geographic region, survey year, and weather conditions during critical growth stages [53] can influence the occurrence of fungi and subsequently mycotoxin spread and concentrations.

In addition to environmental shifts, agricultural practices significantly influence the growth of mycotoxigenic fungi and the subsequent accumulation of mycotoxins in crops, both pre- and post harvest. Poor crop rotation and continuous monoculture, especially in cereals like maize and wheat, create an environment where fungal spores, particularly *Fusarium* species, can persist and build up in the soil, increasing the risk of infection in subsequent planting cycles [54]. Delayed harvesting is another critical factor; crops left in the field beyond maturity are more likely to experience moisture exposure, physical damage, and insect infestation, all of which compromise kernel integrity and promote fungal colonization [55]. Pest damage not only facilitates fungal entry through wounded tissues but also elevates crop stress levels, which can further stimulate the plant’s susceptibility to infection [56]. Similarly, inadequate weed management may allow alternate fungal hosts to survive and transfer spores to the main crop. Irrigation practices also matter, overhead or excessive late-season watering increases ambient humidity around crops, fostering the ideal conditions for aflatoxigenic fungi such as *Aspergillus flavus* [57]. Moreover, the use of contaminated harvesting equipment or unclean storage facilities introduces additional fungal inocula. The use of crop varieties that lack genetic resistance to fungal infection and the overapplication of nitrogen fertilizers, which promote dense, moisture-retaining canopies, further contribute to the risk of contamination [54].

Also, the storage of agricultural produce provides a favourable environment for the growth of toxigenic fungi and the subsequent production of mycotoxins, especially when conditions such as moisture, temperature, aeration, and substrate integrity are poorly controlled. One of the most critical factors is moisture, as fungi require a certain level of water activity (aw) for germination and metabolic activity. When stored grains exceed 13–14% moisture content, they often reach the aw threshold (>0.82) needed for species like *Aspergillus flavus* and *Fusarium verticillioides* to colonize and begin producing mycotoxins such as aflatoxins and fumonisins [58]. In conjunction with moisture, temperature plays a synergistic role; most storage fungi are mesophilic, with optimal growth and mycotoxin biosynthesis occurring between 25 °C and 35 °C. At these temperatures, enzymatic activity is enhanced, allowing fungi to break down the host tissues more efficiently [59]. Furthermore, poor aeration and lack of ventilation in storage facilities create microenvironments rich in carbon dioxide and heat, which not only increase relative humidity but also stimulate the expression of mycotoxin biosynthetic gene clusters in fungi [60]. Compounding these factors, the physical condition of the stored commodity also influences susceptibility to fungal invasion. Grains or nuts that are physically damaged—due to mechanical harvesting, insect infestation, or rodents—provide easier access for fungal hyphae and expose nutrient-rich internal tissues. These damaged substrates are particularly vulnerable if post harvest drying is delayed or uneven, allowing fungi that may have initiated infection in the field, like *Fusarium* spp., to continue colonization in storage [61]. Suboptimal practices in the handling of grains, packaging, and transportation can also exacerbate the risk of contamination [62].

Another critical factor is the type and composition of the raw feed materials. Cereal grains and their by-products are the main components in both finished (complete) and complementary feeds. Finished feed, also known as total mixed ration (TMR), is a nutritionally balanced mixture formulated to meet the full dietary requirements of animals, minimizing the risk of feed separation and selective intake. In contrast, complementary feed provides only part of the daily nutrient intake and must be supplemented with other sources. Since finished feeds are composed of diverse ingredients, each with distinct production and storage conditions, the overall risk of fungal and mycotoxin contamination increases. This is particularly concerning in cereal-based feed production, where mycotoxin levels can reach 60–80%, often surpassing levels found in food crops [55].

Lastly, the association between fungal species and specific cereals significantly influences the distribution of mycotoxins. For instance, *F. verticillioides*, a well-known fumonisin producer, frequently infects maize. Similarly, *F. culmorum* and *F. graminearum*, producers of DON and ZEN, are commonly found in maize, wheat, barley, and rice [63]. Figure 1 depicts a summary of the factors that influence mycotoxin distribution in feeds and feedstuffs.

#### 2.3.2. Global Occurrence of Multi-Mycotoxin in Feed

During feed manufacturing, multiple batches of raw materials, each potentially harbouring different fungal contaminants, are blended to form a complex matrix with a unique mycotoxin risk profile. Given that most feed ingredients are naturally colonized by more than one toxigenic fungus, the co-occurrence of multiple mycotoxins in livestock feeds is a widespread and well-documented phenomenon.

A growing body of evidence underscores the prevalence of multi-mycotoxin contamination. For example, ref. [64] analyzed dairy cattle and poultry feeds in Kenya and found that 96% of the samples contained more than one EU-regulated mycotoxin. Among aflatoxin-positive samples, 100% also contained ZEN, 98% had FUM, 92% contained nivalenol (NIV), and 89% had DON, while smaller proportions included T-2 toxin (6%) and HT-2 toxin (4%). Additionally, 25% of the FB_1_–contaminated samples also contained OTA. Similar findings have been reported globally. In Pakistan, ref. [65] detected an average of 14 different mycotoxins per sample in total mixed rations (TMR) collected from dairy farms in Punjab. In northern Spain, a study by [66] analyzed 400 feed samples across multiple livestock species and found that 63.5% contained at least two mycotoxins. The highest co-occurrence rate was found in poultry feed (70%), followed by sheep (63%), cattle (62%), and pigs (59%). In another Spanish study, ref. [67] reported that 98.7% of pig feed samples were contaminated with multiple mycotoxins, with some samples containing as many as eight different toxins.

On a global scale, multi-mycotoxin contamination remains a significant concern. Ref. [4] reported that 64% of feed samples worldwide contained at least two different mycotoxins, while [68] documented co-contamination in 60% of global samples. Table 3 provides the prevalence of multi-mycotoxin concentrations in finished feed for livestock.

Despite the mounting evidence, the true extent of mycotoxin co-occurrence may still be underestimated. Many studies do not screen for the full spectrum of known mycotoxins, and the limitations in analytical techniques may hinder the detection of all contaminants present in a given sample [80]. Furthermore, although the toxicological interactions between mycotoxins, whether additive, synergistic, or antagonistic, pose significant risks to animal health and productivity, no international regulations currently address the combined effects of multiple mycotoxins [81]. While scientific interest in the biological impacts of mycotoxin mixtures has increased, research in this area remains in its early stages. Therefore, the comprehensive monitoring of mycotoxin co-occurrence is critical, not only to identify the most prevalent combinations but also to inform regulatory priorities and mitigation strategies aimed at ensuring feed safety.

### 2.4. Common Mycotoxin Combinations in Livestock Feed

Recent multi-mycotoxin surveys confirm that mycotoxins rarely occur in isolation; instead, co-contamination of feed materials with multiple mycotoxins is the norm rather than the exception.

While scientific literature extensively documents the effects of individual mycotoxins on various animal species, livestock are more likely to be exposed to multiple mycotoxins simultaneously. For instance, aflatoxins frequently co-occur with FB_1_, while DON, other trichothecenes, and ZEN are commonly found together in the same grain [82]. Among mycotoxin mixtures, FUMs, DON, and ZEN frequently co-occur since they are produced by the same *Fusarium* fungal species [83].

Ref. [66] analyzed 400 compounded feed samples and found that ZEN and DON were the most recurrent combination (23.8%), with 63.5% of the total feed samples containing detectable levels of 2–5 mycotoxins. Similarly, ref. [84] reported that 16.7% of cattle feed samples contained both ZEN (88.2 µg/kg) and DON (289.9 µg/kg) at quantifiable levels. Further evidence of DON and ZEN co-occurrence was reported by [74], who found that these mycotoxins were simultaneously present in 97.8%, 96.7%, and 100% of complete pig feed samples tested in 2018, 2019, and 2020, respectively. In poultry feed, DON and ZEN were detected together in 98.8%, 100%, and 100% of samples from the same years. In Kenya, ref. [78] reported a 35% co-occurrence rate of DON and ZEN in fish feed samples.

AFs and FUMs also co-occur frequently, particularly in sub-Saharan Africa. In addition to their association with FUMs, AFs often appear alongside ZEN and DON in feed materials [82]. A ten-year global survey on mycotoxin prevalence in finished feed found that DON, FUMs, and ZEN were the most common combinations [4]. DON plus ZEN and DON plus FUMs were detected in 48% of feed samples, while ZEN plus FUMs had a co-occurrence rate of 43%.

In China, mycotoxin contamination was equally widespread. A study by [85] found that a combination of AFB_1_, OTA, and/or ZEN was present in up to 100% of 34 feed raw materials and compounded feed samples for cows collected from 18 provinces in 2009. These findings highlight the widespread and complex nature of mycotoxin co-occurrence, emphasizing the need for comprehensive monitoring and mitigation strategies in animal feed production.

## 3. Mechanisms of Mycotoxin Action

### 3.1. Cellular and Metabolic Effects of Mycotoxins

Mycotoxins exert diverse toxic effects on host cells, including oxidative stress, apoptosis, cell death, DNA damage, and cell cycle arrest. The severity of these effects depends on the livestock species and the type of mycotoxin and its metabolites [86]. Ruminants, for instance, exhibit greater resistance to certain mycotoxins compared with monogastric animals, likely due to the detoxifying capabilities of the rumen. Rumen microbes biotransform toxins into more hydrophilic and less toxic metabolites, such as deepoxytrichothecenes, ochratoxin α, aflatoxicol, and glutathione-conjugated AFB_1_ [87]. For example, DON is detoxified in the rumen through epoxide ring opening, which significantly reduces its toxicity by preventing protein synthesis inhibition [88]. In contrast, monogastric animals primarily metabolize mycotoxins through oxidative reactions involving cytochrome P450 enzymes, often resulting in more toxic metabolites [39]. While some mycotoxins share common cellular toxicity pathways, others act through distinct organ systems and biological mechanisms.

AFB_1_ Toxicity

In the liver, cytochrome P450 converts aflatoxins into the highly reactive AFB_1_-8,9-epoxide (AFBO), which binds to DNA and proteins, causing cytotoxicity, DNA damage, and apoptosis [89]. AFBO forms DNA adducts at the N7 position of guanine, leading to GC-to-TA transversions and carcinogenesis [39]. Furthermore, a novel oxidative DNA damage marker, 8-hydroxy-2′-deoxyguanosine (8-OHdG), has been identified in mice, indicating AFB_1_-induced S-phase arrest and carcinogenicity [90]. The conjugation of aflatoxins with glutathione, facilitated by glutathione S-transferase (GST), aids in their detoxification and excretion, but variations in GST and CYP450 enzyme levels influence species-specific susceptibility to aflatoxins [91].

OTA Toxicity

OTA is classified as a category 2B carcinogen due to its potential role in mitochondrial damage, oxidative stress, lipid peroxidation, and apoptosis in various animal models, including pigs [92]. Structurally resembling phenylalanine, OTA disrupts phenylalanine-dependent enzyme activity, impairing gluconeogenesis and inducing cell apoptosis. Additionally, OTA generates high levels of reactive oxygen species (ROS), leading to lipid peroxidation, calcium dysregulation, membrane damage, and DNA strand breaks [93].

FUMs Toxicity

FUMs, particularly FB_1_, FB_2_, and FB_3_, structurally resemble sphinganine, a key precursor of sphingolipids. Their toxic effects stem from their ability to inhibit ceramide synthase, an enzyme essential for sphingolipid metabolism and cellular homeostasis. This inhibition results in the accumulation of sphinganine and other sphingoids, which disrupt cellular functions, leading to oxidative stress, apoptosis, and abnormal cell differentiation [94]. Fumonisin exposure has been linked to leukoencephalomalacia in horses, pulmonary oedema, reduced weight gain, and liver damage in swine [95].

T-2 Toxin and DON Toxicity

Trichothecenes are sesquiterpenoid mycotoxins classified into four types (A–D) based on their functional hydroxyl and acetoxy side groups. Type A (e.g., T-2 and HT-2 toxins) and type B (e.g., DON) are the most toxic and widely distributed. Trichothecenes inhibit eukaryotic protein synthesis by binding to the 60S ribosomal subunit and interfering with peptidyl transferase activity [96]. This disruption triggers phosphokinase stress responses, leading to proinflammatory gene activation, gastrointestinal impairment, and eventual cell death [97]. Proteomic analysis of T-2 toxin-exposed porcine hepatocytes revealed significant upregulation of mitochondrial proteins, highlighting the toxin’s role in metabolic dysfunction [98].

DON specifically binds to ribosomes, inhibiting translation and activating the ribotoxic stress response (RSR). DON exposure leads to the phosphorylation of multiple proteins involved in protein synthesis and signal transduction, suggesting a broader regulatory role in immune function [99]. Further studies revealed that DON significantly alters the phosphorylation of 188 proteins, indicating that its impact on ribotoxic stress extends beyond translation inhibition and MAPK activation [100].

ZEN Toxicity

ZEN primarily exerts its toxic effects by inducing oxidative stress, DNA fragmentation, and chromosomal aberrations in bone marrow cells [101]. As a mycoestrogen, ZEN stimulates estrogen receptor-mediated cellular proliferation in mammary gland cells and promotes the growth of MCF-7 breast cancer cells by inhibiting apoptosis via Bax/Bcl-2 regulation [102]. In ZEN-treated HepG2 cells, proteomic analysis has identified 99 differentially expressed proteins involved in cellular maintenance, cancer progression, molecular transport, and carbohydrate metabolism [103]. The pathway analysis linked these alterations to disruptions in reproductive health, cell morphology, and metabolic function.

### 3.2. Toxicological Interaction of Mycotoxins in Animal Production

In animal production, the clinical signs of mycotoxicosis observed in the field often cannot be explained by the low levels of individual mycotoxins detected in feed. This discrepancy is largely due to the combined effects of multiple mycotoxins, which can be significantly greater than their individual toxicities [104]. Despite the critical importance of understanding mycotoxin interactions, the literature on this subject remains limited. A major constraint in studying mycotoxin interactions in animal production is that most research is conducted in vitro using cellular models rather than in vivo animal models. However, the in vivo toxicity of mycotoxin interactions cannot be accurately predicted from in vitro studies, as the complexity of multiple mycotoxin interactions leads to diverse effects, including additive and synergistic outcomes. The toxic response is further influenced by variables such as cell type, exposure duration, mycotoxin concentration, and the experimental models used [105].

#### 3.2.1. Types of Mycotoxin Interactions

When livestock are exposed to multiple mycotoxins simultaneously, their combined effects can be synergistic, additive, or antagonistic, depending on how the toxins interact within the animal’s system.

Synergism occurs when the toxic effect of a mycotoxin combination exceeds the sum of their individual toxicities [106]. This is distinct from potentiation, where one or more mycotoxins, non-toxic or minimally toxic on their own, induce significant toxicity when combined [107]. Synergistic interactions can greatly enhance overall toxicity. For instance, co-exposure to AFs and OTA or T-2 toxin leads to intensified toxic effects [108]. FB_1_ has been shown to enhance aflatoxin absorption and carry over into milk, compounding health risks [109]. Even at low to moderate concentrations, OTA (101.41 µg/kg) interacts synergistically with AFB_1_ (20.10 µg/kg) to exacerbate nephropathy in chickens [110]. Diets co-contaminated with AFB_1_ and OTA resulted in higher concentrations of AFM_1_ in tissues and greater renal toxicity than diets containing AFB_1_ and ZEN [111]. Furthermore, emerging mycotoxins can intensify the effects of regulated ones, for example, DON with fusaric acid, or diacetoxyscirpenol with AFs [112].

Additive interactions occur when the total toxic effect of co-exposure is equivalent to the sum of individual toxin effects. A variation on this, termed “less than additive”, describes scenarios where the combined toxicity does not exceed that of the most toxic mycotoxin alone [107]. Notable additive combinations include AFB_1_ with OTA, T-2 toxin, FB1, or DON; FB1 with T-2 toxin; and OTA with T-2 toxin. In poultry, DON and AFs show additive toxicity [113]. Emerging mycotoxins, such as moniliformin, also exhibit additive interactions with fumonisin or DON [114]. Among *Fusarium*-derived toxins, interactions may range from additive to synergistic, impacting animal performance indicators such as feed intake, weight gain, and mortality.

Antagonism occurs when the combined effect of multiple mycotoxins is less than the sum of their individual toxicities [107]. In this case, one mycotoxin may reduce or negate the toxicity of another. For example, the combination of DON and ZEN has shown antagonistic effects on the immune function of pigs. In one mouse study, antioxidant capacity and malondialdehyde levels, markers of oxidative stress, were lower in the group exposed to both DON and ZEN than in those exposed to either toxin individually, suggesting an antagonistic effect [115]. In vitro studies also show reduced stress responses when DON and ZEN are combined, possibly due to hepatocyte compensatory mechanisms [116].

#### 3.2.2. Factors Influencing Mycotoxin Interactions

The nature of mycotoxin interactions, synergistic, additive, or antagonistic, is influenced by various biological and experimental factors. These include the specific toxicological endpoint being evaluated, the animal species, age, sex, nutritional status, mycotoxin dose, as well as the duration and route of exposure [104].

For instance, studies using HepaRG human hepatic cells demonstrated that short-term co-exposure (3 and 12 h) to DON and ZEN significantly increased cell mortality compared with individual exposures, indicating a synergistic effect. However, at 18 h, ZEN alone induced apoptosis and necrosis, while DON had no significant cytotoxic effect. Interestingly, the combined exposure at this point mimicked the effects of ZEN alone, suggesting a less-than-additive interaction [117].

Immune cell responses to DON and ZEN vary by cell type and experimental conditions. In porcine splenic lymphocytes, both mycotoxins independently triggered oxidative stress and apoptosis in a dose-dependent manner, but their combination produced a synergistic cytotoxic response [118]. Similarly, in lymphocytes isolated from the venae cava cranialis of pigs, DON and ZEN each exhibited genotoxic effects; however, their interaction was antagonistic at low doses and synergistic at higher doses, particularly when assessed after 72 h of exposure [119].

Further in vivo evidence comes from a study on piglets that were fed a controlled diet for three weeks. Neither DON nor ZEN alone significantly affected body weight gain or average daily feed intake. However, piglets co-exposed to both toxins showed a marked reduction in these parameters, indicating synergistic effects on intestinal function and systemic inflammation [120].

These findings highlight the potential for toxicological interactions to exacerbate the harmful effects of regulated mycotoxins, even when individual concentrations remain within legal limits. Therefore, the presence of multiple mycotoxins in animal feed constitutes a substantial hazard to both animal and public health. As such, routine monitoring and risk assessment of co-occurring mycotoxins are essential for safeguarding feed quality and ensuring food safety.

## 4. Health Impacts on Livestock

Poor livestock performance and/or disease symptoms observed in commercial operations may result from the interactions between multiple mycotoxins. The enhanced bio-transfer of mycotoxins from feed to animal tissues and animal-source foods such as eggs and milk when exposed to mycotoxin mixtures is of greater concern. For instance, the toxicological interactions between AFs and ZEN or OTA have led to an increased accumulation of ZEN and OTA residues in the organs of broiler chickens compared with those fed individual mycotoxins [121].

### 4.1. Effect on Growth Performance

While much of the early research has focused on the effects of individual mycotoxins, increasing evidence shows that livestock are commonly exposed to multiple mycotoxins simultaneously under field conditions [4,64]. Among the most concerning outcomes of such interactions is the depression in growth performance, a key productivity indicator in animal agriculture. Mycotoxin interactions can compromise nutrient utilization, reduce feed intake, induce organ damage, and impair metabolism, ultimately leading to reduced weight gain and feed efficiency [120,121]. These effects are especially pronounced in monogastric species like pigs and poultry, which lack the ruminal detoxification mechanisms present in ruminants [122].

Although ruminants generally exhibit greater resistance to individual mycotoxins due to microbial detoxification in the rumen, multiple mycotoxin exposures can still significantly impair performance, especially in young or high-producing animals. For example, ref. [123] demonstrated that replacement heifers fed a diet containing AFB_1_ (>10 μg/kg) and FB_1_ + FB_2_ (5000–20,000 μg/kg) experienced delayed growth. Similarly, ref. [124] reported that while AFB_1_ alone or in combination with ZEN had a minimal impact, the combination of AFB_1_ (50 μg/kg), OTA (100 μg/kg), and ZEN (500 μg/kg) significantly reduced dry matter intake (DMI) in dairy goats. Ref. [125] similarly reported that lactating cows consuming a TMR contaminated with multiple low-level mycotoxins (including AFB_1_ (38 μg/kg), ZEN (541 μg/kg), OTA (501 μg/kg), T-2 (270 μg/kg), DON (720 μg/kg), and FB_1_ (701 μg/kg)) experienced significant reductions in DMI and milk production, effects not observed when animals were exposed to individual toxins at the same concentrations. These findings emphasize that even subclinical levels of multiple mycotoxins can lead to compounded stress on metabolic and immune functions, indirectly impairing growth and production. Other studies have shown that individual exposure to mycotoxins such as ZEA (500 mg/kg) or OTA (3.5 mg/kg) produced no significant effect on growth in cows and sheep [126,127], yet the impact of these toxins escalated markedly when present in combination [128].

The situation is even more critical in pigs, which are highly sensitive to *Fusarium*-derived mycotoxins. Ref. [120] found that neither DON (1000.6 μg/kg) nor ZEN (269.1 μg/kg) alone significantly affected piglets’ feed intake or body weight gain, but their combination resulted in substantial reductions in both parameters. This synergistic effect is attributed to exacerbated intestinal inflammation and systemic toxicity, which would not occur under individual exposures. Additionally, DON and ZEN co-exposure has shown antagonistic effects on immune response but synergistic impairments in gastrointestinal and reproductive function. In breeding pigs, combined mycotoxin exposure has been associated with reduced litter size and sow productivity due to impaired nutrient assimilation and elevated liver enzyme activity [129].

In poultry, the effects of co-exposure to multiple mycotoxins are diverse, with significant influences on growth performance, organ health, and immune function. Several studies have shown that while individual mycotoxins may cause moderate reductions in body weight gain, their combinations can significantly amplify these effects. For instance, ref. [130] observed that turkey poults fed diets containing combinations of FB_1_ (300 mg/kg) with either DAS (4 mg/kg) or OTA (3 mg/kg) experienced 46% and 37% reductions in body weight gain, respectively, which were much greater than the 8–30% reductions caused by the individual toxins. Similarly, a combination of FB_1_ (200 mg/kg) and AF (0.75 mg/kg) resulted in a 47% reduction in weight gain, exceeding the effects of FB_1_ (10%) and AF (39%) alone, suggesting an additive or slightly enhanced toxic interaction [131]. In broiler chickens, ref. [113] reported that the co-administration of AF (2.5 µg/kg) and DON (16 µg/kg) led to additive toxicity across multiple performance and health parameters. These additive or less-than-additive effects indicate that the impact of combined mycotoxins may equal or slightly exceed the sum of individual effects. However, more potent synergistic effects have been documented in cases where AF was combined with OTA [113,132,133], T-2 toxin [134], or DAS [135], leading to disproportionately severe reductions in growth and increased organ damage. Conversely, some combinations can result in antagonism; [136] found that co-exposure to OTA (2 mg/kg) and DON (16 mg/kg) in growing broiler chicks led to a lower-than-expected toxic response, with less damage compared with the individual toxin effects, indicating a protective or inhibitory interaction between the two. Importantly, these interactions are often associated with increased histopathological damage to the gastrointestinal tract, which impairs nutrient absorption and further reduces feed efficiency and growth.

The detrimental effects of mycotoxin interactions are not limited to traditional livestock. Rabbits, due to their monogastric digestive systems, are also vulnerable. Ref. [137] showed that co-exposure to AFB_1_ (0.5 ppm) and OTA (1 ppm) caused a 40% reduction in body weight gain and 25% mortality, compared with a 12% weight reduction and 12.5% mortality when each toxin was administered separately. According to the authors, mortality was 15% for the FB_1_ + T-2 combinations and 0 in the control group, with the overall results indicating additive toxicity with FB_1_ + T-2 and less-than-additive toxicity with FB_1_ + DON combinations. These results demonstrate a clear synergistic effect on growth performance and survival. Ref. [137] further linked these effects to increased oxidative stress, decreased protein synthesis, and liver damage. The diversion of nutrients toward repairing damaged organs, rather than growth, was identified as a major contributor to performance suppression. Table 4 summarizes the effect of multi-mycotoxins on the growth performance of different livestock species.

### 4.2. Immune Function, Antioxidant Status, and Reproductive Effects

Mycotoxins compromise immune function by modulating cytokine production, suppressing immunoglobulin synthesis, and impairing the activity of immune cells. These immunosuppressive effects are magnified when mycotoxins are present in combination, posing greater risks to animal health and productivity than single toxin exposures. In pigs, a species particularly sensitive to *Fusarium*-derived toxins, combined mycotoxin exposure has been shown to cause profound immune suppression. Ref. [138] demonstrated that pigs exposed to a chronic combination of AFB_1_ (60, 120, and 180 µg/kg) and DON (300, 600, and 900 µg/kg) exhibited markedly synergistic reduced lymphocyte proliferation, decreased serum immunoglobulins (IgG, IgA), and downregulated cytokine expression (IL-2, IFN-γ), compared with single toxin exposure. According to the authors, concentrations of 120 μg of AF/kg and 600 μg of DON/kg or above result in severe altered immune health, systemic inflammation, and partial liver damage, causing further reduction in the growth of pigs.

Beyond immune dysfunction, histopathological findings also reflect the complexity of mycotoxin interactions at the organ level. Ref. [139] observed that pigs fed diets co-contaminated with AFB_1_ (0.375/0.75 mg/kg) and OTA (1/2 mg/kg) developed hepatic lesions similar to those fed AFB_1_ alone. Interestingly, renal lesions were less severe and associated with lower creatinine and blood urea nitrogen concentrations in co-contaminated pigs compared with those fed OTA alone, suggesting that the toxic effects of combined mycotoxins may vary by organ system and are not always additive, highlighting the need for organ-specific assessments in evaluating multi-mycotoxin toxicity.

In addition, reproductive toxicity in pigs is also a major concern. Ref. [140] reported that that co-exposure of oocytes to ZEN (3.12 μmol/L) and DON (3.12 μmol/L) induced aneuploidy and abnormal egg development in pigs, an activity attributed to the synergistic suppression of ovarian steroidogenesis. A reduced expression of aromatase and LH receptor mRNA was also observed, effects not present when oocytes were exposed to individual mycotoxins. These findings highlight that reproductive impairments from multiple mycotoxins may be more severe than initially assumed based on single toxin studies.

In poultry, combined mycotoxin exposure similarly leads to exacerbated health problems. Ref. [141] indicated that a high-dose combination of AF (0.5, 1, and 2 ppm) and OTA (1, 2, and 4 ppm) in diets fed to broiler chickens severely suppressed cell-mediated immunity and haemagglutination titers, underlining the additive and synergistic nature of these interactions. Histopathological outcomes are equally concerning; ref. [142] found that chickens fed a co-contaminated diet of AFB_1_ (0.2 ppm) and OTA (0.2 ppm) developed more severe hepatic lesions, including granular degeneration, necrosis of liver parenchyma, and hemorrhages, compared with single toxin diets. Similarly, renal injuries appeared earlier and were more pronounced in birds exposed to multiple mycotoxins, characterized by the destruction of the tubular epithelium and the detachment of tubular cells from the basement membrane.

The interaction between AFB_1_ and FB_1_ in broilers further highlights the complexity of combined mycotoxin effects. Ref. [143] reported that broilers receiving diets with both AFB_1_ (0, 50, and 200 μg/kg) and FB_1_ (0, 50, and 200 mg/kg) from 8 to 41 days of age exhibited severe liver damage, bile duct proliferation, and trabecular liver disorders, along with elevated plasma total protein concentrations. While AFB_1_ alone caused notable pathology and FB_1_ caused milder changes, their combination induced the most pronounced lesions, suggesting a synergistic or additive effect. In their earlier study examining the impact of the same mycotoxin combination and concentration on body weight, antibody titres, and the histology of broiler chicks, ref. [144] observed that the combined toxins had primarily additive effects on body weight, liver structure, and the immunological response of broilers at the concentration used. Specifically, the main histological alterations included significant vacuolar degeneration and cell proliferation of the bile ducts in the liver, and hydropic degeneration in renal tubules in the kidneys. Immunological impacts varied across studies; ref. [131] observed a less-than-additive reduction in lymphocyte proliferation with AF and FB_1_ co-exposure. Ref. [143] noted a synergistic decrease in Newcastle disease antibody titers, although ref. [145] observed an unexpected increase in hemagglutination titers against sheep red blood cells in turkey poults. These inconsistencies point to the complexity of assessing immune responses under co-exposure scenarios.

The underlying mechanism for these immune impairments frequently involves oxidative stress. Ref. [146] proposed that mycotoxin-induced oxidative imbalance results in lipid peroxidation and the depletion of antioxidant enzymes, leading to cellular dysfunction and liver damage. This was confirmed by [147] in broilers fed naturally contaminated diets containing multiple mycotoxins (AFs, ZEN, FBs, and DON), which showed elevated oxidative stress markers and compromised antioxidant defences.

Although ruminants are generally considered more resilient to mycotoxin toxicity due to microbial detoxification in the rumen, co-exposure can still result in significant health challenges, particularly in young or high-producing animals. Ref. [148] reported that calves fed DON and ZEN displayed reduced white blood cell counts and impaired neutrophil function, although the effects were somewhat attenuated compared with monogastric species. However, more severe impacts have been documented in dairy goats. Ref. [122] showed that simultaneous exposure to AFB_1_ (50 μg/kg), OTA (100 μg/kg), and ZEA (500 μg/kg) significantly impaired hematological parameters and elevated liver enzyme levels (ALT and ALP), effects absent with individual toxin exposures, highlighting an additive to synergistic interaction.

In dairy cows, similar findings were observed by [149], where co-contamination with ZEA (625 μg/kg), DON (471 μg/kg), and AFB_1_ (21.2 μg/kg) negatively affected blood parameters, increased somatic cell counts, and impaired immune function. Also, ref. [150] showed that cows consuming *Fusarium*-contaminated TMR containing DON (3.2 mg/kg), 15-Acetyl-deoxynivalenol (0.28 mg/kg), and ZEN (0.24 mg/kg) exhibited elevated total serum protein, globulin, urea, sodium concentrations, and osmolality levels. The fungal origin of the mycotoxins also appears to influence toxicity severity; ref. [26] emphasized that mycotoxins produced by the same fungal genus, such as AFB_1_ and OTA from *Aspergillus*, often exert stronger combined effects. Supporting this, ref. [122] found that the AFB1 + OTA combination produced greater elevations in liver enzymes and oxidative stress markers (MDA) while reducing antioxidant enzyme activities (SOD and GSH-Px) more significantly than AFB_1_ + ZEA. This cumulative oxidative burden was further highlighted by [122], who demonstrated that dairy goats fed a combination of AFB_1_, OTA, and FB_1_ experienced a synergistic increase in lipid peroxidation and a severe depletion of antioxidant enzymes, reinforcing the substantial threat posed by multi-mycotoxin contamination even in ruminants.

Rabbits, too, suffer serious consequences from combined mycotoxin exposure. Based on the haematotoxic study by [151], the combined effect of FB_1_ (10 mg/kg) and T-2 (2 mg/kg) on weaned rabbits exhibited an antagonistic effect on the red blood cell NA(+)/K(+) ATPase activity, while the activity decreased in the T-2 group and increased in the FB1 group. Ref. [137] documented additive liver damage and increased mortality with AFB_1_ + OTA co-exposure, where a 25% mortality rate was recorded, double the rate seen in single toxin groups. These experimental outcomes align with spontaneous outbreaks of aflatoxicosis and ochratoxicosis in rabbits, such as the 75% mortality recorded in Himachal Pradesh, India [152]. The co-exposure of the animals to multiple mycotoxins, including T-2 toxin and DON, lead to elevated ROS generation, DNA fragmentation in lymphoid tissues, and reduced total antioxidant capacity, with more pronounced damage than seen with single toxin exposures. For instance, a study by [153] investigated the effects of these mycotoxins on immune cells and found that co-exposure led to increased oxidative stress and apoptosis in RAW264.7 cells. The study concluded that the JAK/STAT pathway plays a critical role in the proinflammatory gene expression and apoptosis induced by these toxins. Table 5 summarizes the effect of multi-mycotoxin on the immune and reproductive indices in different livestock species.

### 4.3. Effect on Production Indices

The profitability of the livestock industry is closely tied to the quantity and quality of animal products such as milk, meat, eggs, wool, and skin, particularly under conditions where production costs are minimized. While ruminants are generally considered less susceptible to the adverse effects of mycotoxins due to microbial detoxification in the rumen, evidence shows that key production indices can still be significantly compromised, especially under co-exposure to multiple mycotoxins.

In pregnant animals, for example, combined mycotoxin exposure has been shown to impact their productive and reproductive performances, including their offspring. Ref. [154] reported that pregnant Ossimi sheep fed a diet contaminated with AFB1 (50 µg/kg DM) and OTA (100 µg/kg DM) showed percentage decreases of 37.5%, 44.6%, 44.12%, 47.93%, 39.29%, and 35.77% for fertility rate, pregnancy rate, lambing rate, litter size, fecundity rate, and stillbirth, respectively. Additionally, the milk compositions (fat, protein, lactose, and total solids) of the Ossimi ewes decreased significantly. Similarly, dairy cows and goats appear highly sensitive to multi-mycotoxin challenges. Complementary findings by [124] demonstrated that dairy goats exposed to different combinations of AFB_1_ (50 µg/kg DM), OTA (100 µg/kg DM), and ZEN (500 µg/kg DM), suffered significant decreases in milk production, whereas AFB_1_ alone had minimal effects, suggesting a synergistic interaction.

In dairy cows, multi-mycotoxin contamination also severely affects lactational performance. For instance, [155] found that feeding lactating cows with mouldy (natural co-contaminations) corn meal (AF: 3 µg/kg; ZEN: 490 µg/kg) and cottonseed (AF: >150 µg/kg; ZEN: 84 µg/kg) led to a decreasing trend for the dry matter intake, milk yield, 4% fat-corrected milk, and energy-corrected milk. The mechanisms involved include direct inhibition of lactogenesis and indirect effects such as reduced dry matter intake (DMI), altered rumen fermentation, and liver dysfunction. Supporting this, [125] observed that a total mixed ration contaminated with a spectrum of mycotoxins (AFB_1_ (38 µg/kg), ZEN (541 µg/kg), OTA (501 µg/kg), T-2 (270 µg/kg), DON (720 µg/kg), and FB_1_ (701 µg/kg)) markedly suppressed milk production. Ref. [156] further highlighted that cows fed impure aflatoxin mixtures produced less milk compared with those exposed to purified toxins, reinforcing the compounded risks of multi-mycotoxin contamination.

Poultry production systems are equally vulnerable to the negative impacts of mycotoxin co-exposure. In the study conducted by [157], a higher concentration of the combinations of DON (0.1–0.9 mg/kg), FUM (1–33 mg/kg), and ZEN (0.4–3.5 mg/kg) significantly reduced the production performance of broiler chickens. In laying hens, co-exposure to mycotoxins results in substantial reductions in egg production and hatchability. Ref. [158] found that the simultaneous exposure of laying hens to AF (0.5–2 ppm) and OTA (1–4 ppm) reduced egg production and feed efficiency, with interactions varying from synergistic to slightly less than additive depending on the concentration. In contrast, [159] reported an antagonistic interaction between AFB_1_ (50–200 µg/kg) and FB_1_ (10 mg/kg) in quails, where the reduction in egg production was less pronounced when both toxins were combined compared with FB_1_ or AFB_1_ alone. Nevertheless, the overall trend across the studies emphasizes the detrimental effect of multi-mycotoxin exposure on laying performance. Ref. [160] further demonstrated that the feeding of grains naturally contaminated with *Fusarium* (DON [11.7 µg/g]; 15-acety-DON [0.4 µg/g], and ZEN0 [0.6 µg/g]) mycotoxins exhibited decreased egg production, egg mass, and egg and eggshell weights, and a surge in unmarketable eggs, raising serious productivity and food safety concerns. A similar lower egg mass was also reported in laying hens fed a diet contaminated with T-2 (4 ppm) and FB_1_ (20 ppm) [161].

In swine production, reproductive performance is particularly sensitive to combined mycotoxin exposure. Co-exposure to ZEN and DON has been shown to disrupt reproductive physiology, impairing herd fertility and piglet survival. Ref. [162] reported that female weaned piglets consuming *Fusarium* toxin-contaminated maize (ZEN-1.2 mg/kg and DON-8.6 mg/kg) diets experienced an increased mean weight (almost 100% compared with the control) of the uterus relative to the body weight as well as a decreased serum concentration of follicle stimulating hormone. These reproductive challenges translate into substantial economic losses for swine producers.

Beyond reproduction, mycotoxin mixtures also impair carcass traits and meat quality in grower–finisher pigs. Ref. [104] demonstrated that pigs exposed to DON, ZEN, and FB_1_ exhibited increased backfat thickness, reduced loin eye area, and inferior meat quality, including darker colouration and lower water holding capacity. These alterations affect marketability and consumer acceptance, further exacerbating the financial burden associated with mycotoxin contamination. Table 6 is a summary of the impact of multiple mycotoxins on the production indices of different livestock species.

## 5. Conclusions

Given the fact that the co-occurrence of mycotoxins is the rule rather than the exception, knowledge about their presence should be intensified, and their combination investigated more closely in finished feed. This could be realized by continuously monitoring their presence in finished feed as well as establishing reliable methods for generating accurate co-occurrence data. Due to their high mycotoxin content, understanding the multi-mycotoxin contaminations in animal feedstuffs and their associated toxicological mechanism remains an important aspect of public health.

The co-occurrence of multiple mycotoxins in livestock feed is no longer an exception but the norm, significantly amplifying health risks and production losses across species. Despite the increasing awareness, regulatory frameworks largely focus on individual mycotoxins, neglecting the additive or synergistic effects that arise from multi-mycotoxin exposures. This review underscores that even sub-threshold levels of individual toxins, when combined, can lead to profound physiological, reproductive, immunological, and performance deficits in livestock, ultimately threatening food security and farm profitability. Continuous monitoring and surveillance of finished feed for multiple mycotoxins, coupled with the development of advanced analytical techniques, are crucial steps forward. Furthermore, future risk assessment models must incorporate cumulative and interactive toxicity rather than relying solely on isolated toxin limits. To effectively address the complex challenges of mycotoxin co-contamination, it is essential to adopt a comprehensive approach that combines better farming practices, new detoxification methods, and updated regulatory standards. Strengthening global collaboration in research, regulation, and education will be vital in ensuring safer feed supplies, sustainable livestock production, and public health protection in an increasingly interconnected world.

## 6. Future Research Direction

Future studies should prioritize comprehensive toxicological investigations of multi-mycotoxin interactions using in vivo models across species, particularly with low dose, chronic exposures that reflect field conditions. Additionally, omics-based approaches (transcriptomics, metabolomics, and proteomics) could uncover underlying cellular and molecular pathways involved in co-toxicity, enabling the development of biomarkers for early detection. There is also a pressing need to explore climate-resilient agricultural and storage practices to limit fungal proliferation and mycotoxin synthesis. Strengthening international surveillance systems and creating data-sharing platforms for co-occurrence patterns would support global food safety and harmonized regulation. By addressing these gaps, future research can contribute to a more holistic, science-driven approach to mitigating mycotoxin risk in livestock production and ensuring safe animal-derived food for human consumption.

## Figures and Tables

**Figure 1 toxins-17-00365-f001:**
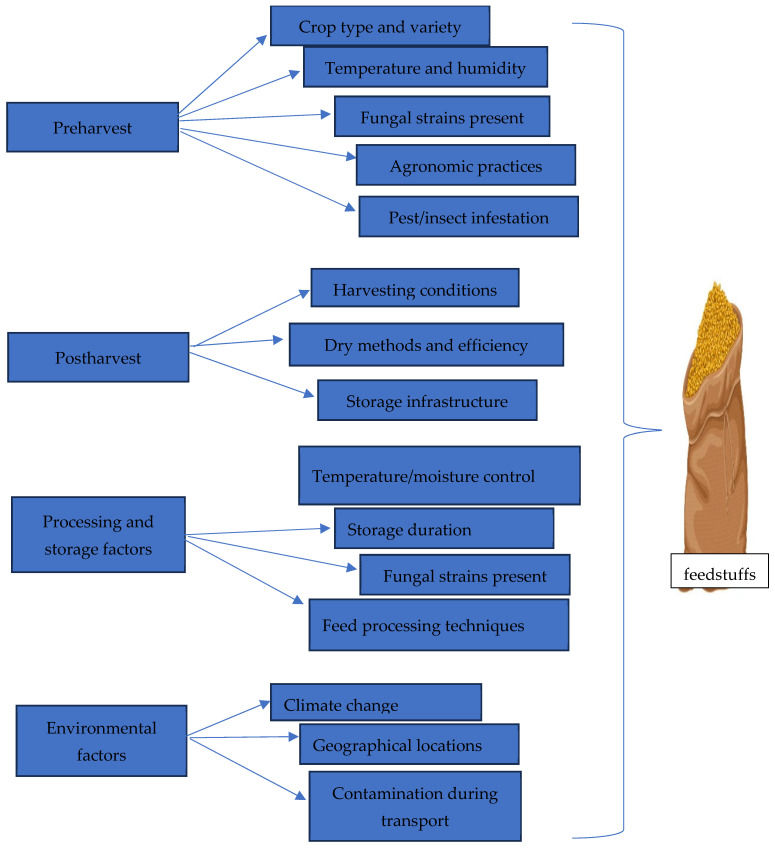
Factors influencing the spread of mycotoxin in feedstuffs.

**Table 1 toxins-17-00365-t001:** Mycotoxin targets and toxicity factors in animals.

Mycotoxins	Targeted Organs	Order of Sensitivity	Age/Sex-Related Physiological Factors
FB_1_	Liver, kidney, and small intestine [34]	Pig > poultry > ruminant > fish [35]	Females exhibit greater sensitivity [35]
AFB_1_	Kidney and liver [36]	Poultry > rabbit > pig > ruminants [37]	Young and female animals have increased susceptibility [38]
OTA	Liver and kidney [39]	Pig > poultry > rabbits > ruminants [39]	Male animals have increased susceptibility [40]
DON	Liver, kidney, and lymphocyte [41]	Pig > poultry > ruminants [41]	Male and older animals exhibit greater sensitivity [41]
T-2	Liver, kidney, and lymphocyte [41]	Poultry > pigs > ruminants [41]	Heightened susceptibility in young animals [41]
ZEN	Reproductive organs (ovaries, uterus, vulva and vagina, mammary glands, testes), liver, and kidney [24]	Pig > dairy cattle > poultry > ruminants [24]	Young animals, particularly prepubertal females, exhibit increased susceptibility [24]

**Table 2 toxins-17-00365-t002:** International maximum limits for major mycotoxins in animal feed.

Mycotoxin	Agency	Species/Feed Type	Maximum Limit (mg/kg)	Notes	References
Aflatoxins (Total/AFB_1_)	EU	All feed materials (AFB_1_)	0.02	Maximum content for AFB_1_ in feed	EC Directive 2002/32/EC
	USA (FDA)	Dairy cattle, breeding animals	0.020	Enforceable legal limit	FDA Compliance Policy Guide Sec. 683.100 (2022)
	USA (FDA)	Finishing cattle, swine, poultry	0.1	Higher threshold allowed	FDA CPG Sec. 683.100 (2022)
	Codex	General feed	0.050	Recommended maximum (not binding)	Codex Alimentarius, CAC/RCP 45-1997
Ochratoxin A (OTA)	EU	Swine feed	0.05	Guidance value	EC Recommendation 2006/576/EC
	EU	Poultry feed	0.1	Guidance value	EC Recommendation 2006/576/EC
	Canada	General feed (guideline)	0.25	Non-binding	CFIA Guidance for OTA in Livestock Feed (2009)
	USA	—	No official limit	Not regulated in feed	Not regulated
Fumonisins (FB_1_ + FB_2_)	EU	All feed materials	5	Applies mainly to maize products	EC Recommendation 2006/576/EC
	USA (FDA)	Swine	10	Swine are highly sensitive	FDA Guidance for Industry #221 (2011)
	USA (FDA)	Poultry	30	Includes broilers and layers	FDA Guidance #221
	USA (FDA)	Cattle	100	Feedlot and breeding cattle	FDA Guidance #221
	Codex	Maize products for feed	60	Proposed guideline	Codex Alimentarius (CAC/RCP 56-2004)
Deoxynivalenol (DON/Vomitoxin)	EU	Pig feed	0.9	Guidance value	EC Recommendation 2006/576/EC
	EU	Poultry, cattle feed	5	Less sensitive species	EC Recommendation 2006/576/EC
	USA (FDA)	Swine feed	1	Advisory level	FDA Guidance for Industry #186 (2010)
	USA (FDA)	Poultry	5		FDA Guidance #186
	USA (FDA)	Ruminants	10	For beef and dairy	FDA Guidance #186
Zearalenone (ZEN)	EU	Piglets, sows	0.1–0.25	Reproductive sensitivity	EC Recommendation 2006/576/EC
	EU	Cattle and poultry	0.5–3	Guidance values by species	EC Recommendation 2006/576/EC
	USA	—	No official federal limit	Occasional guidance by states or industry	Not regulated
T-2/HT-2 Toxin	EU	Cereal-based feed	0.25–0.5	Guidance only	EC Recommendation 2013/165/EU
	Canada	General feed (guideline)	0.1–1	Based on species sensitivity	CFIA Feed Contaminants Guidelines
	USA	—	No limit	Not officially regulated	Not regulated

**Table 3 toxins-17-00365-t003:** Prevalence of multi-mycotoxin concentrations in finished feed for livestock.

Country	Feed Types	Number of Samples Analyzed	% Containing Two or More	Mycotoxins Analyzed	Highest Mycotoxin Combinations	Reference
Brazil	Cattle feed and ingredients	1329	87% (≥2); 28.6% (=3); 22.5% (=4); 11.4% (=5); 1.16% (=6)	AFs, DON, FUMs, OTA, T-2, ZEN	DON + ZEN-45.2%; AF + DON-42.1%; AF + ZEN-41.5%	[69]
South Africa	Dairy cattle feed and forages	300–600	66% (≥2); 20% (=2)	AFs, DON [+3-DON, 15-ADON], ZEN, FUMs, OTA, T-2 toxin (T-2) and HT-2 toxin (HT-2), NIV, DAS, FUS-X, NEO, AOH, AME, ROQ-C, ENN B, STERIG	DON + FUM + ENN B	[70]
	Poultry feeds	105	100%	ZEN + metabolites, T-2, FUMs, AFs, HT-2, AME, DON, 3-ADON, 15-ADON	AFs + FUMs + ZENs + DON-42%	[71]
	Compounded feed for all classes of livestock	92	98.9%	DON, ZEN, FUM, OTA, AF, T-2/HT-2	DON + ZEA-99%; FB + DON + ZEN-67%; FB + DON + ZEN + AF-26% FB + DON + ZEN + AF + OTA-5.5%	[5]
Global multi-country survey	Feed and feed raw materials	74,821	64%	ZEN, DON, FUMs, OTA, T-2	DON + ZEN + FUMs	[4]
	Feedstuffs and feed	7049	48% (≥2)	DON, ZEN, FUM, OTA, AF, T-2/HT-2	Not stated	[72]
China	Feed ingredients and pig finished feed	1569 (742 feed ingredients; 827 finished pig feed)	100%	AFB_1_, ZEN, DON	Not stated	[73]
	Feed samples	3507 (1417 complete/finished feed; 2083 feedstuffs)	~100% (finished feed)	AFB_1_, ZEN, DON	AFB_1_ + DON (99.6%-pig feed; 99.7%-poultry feed; 99.3%-ruminant feed); AFB_1_ + ZEN (99.5%-pig feed; 99.7%-poultry feed; 99.3%-ruminant feed); DON + ZEN (98.2%-pig feed; 99.6%-poultry feed; 98.6%-ruminant feed); AFB_1_ + DON + ZEN (99.1%-pig feed; 99.6%-poultry feed; 98.6%-ruminant feed)	[74]
Poland	Feed materials and feedstuffs	3980 (642 maize, 2027 feed samples, 990 small grains, 142 maize silage, and 179 TMR samples)		AFs, FUMs, ZEN, OTA, T-2, HT-2, DON	DON + ZEN (98.7%-complete feed, 100%-TMR), DON + T-2 + HT-2 (97.7%-complete feed, 97.2%-TMR), DON + T-2 + HT-2 + ZEN (89.3%-complete feed, 97.2%-TMR), ZEN + T-2 + HT-2 (89.4%-complete feed, 97.2%-TMR)	[75]
Thailand	Dairy feed samples	115	96.6% (≥2)	69 metabolites, including major mycotoxins	ZEN + FB_1_-65.9%, ZEN + DON-56.8%	[76]
Pakistan	Poultry feeds	150	100%	AFs, DON, NIV, ZEN, NEO, OTA T-2, HT-2, 3-ADON, DAS, 15-ADON, STC, DOM-1, F-X	AFs + FBs-100%	[77]
Spain	Compounded feed for cattle, pigs, poultry, and sheep	400	63.5% (≥2); 37.8% (=2); 16.8% (=3); 7.3% (=4); 1.5% (=5)	AFs (B_1_, B_2_, G_1_, G_2_), OT (A, B), ZEN, DON, STER	ZEN + DON (23.8%); AFG_2_ + ZEN + DON (13%) and AFB1 + ZEN + DON (11%); AFG_2_ + AFG_1_ + ZEN + DON (2%); AFB_2_ + AFB_1_ + ZEN + DON + STER (3%)	[66]
Kenya	Fish feed	78	87% (≥2); 13% (=8); 1% (=17)	Regulated (DON, ZEN, FUM, OTA, AF, T-2/HT-2) and non-regulated mycotoxins	Not stated	[78]
	Dairy and poultry feed	67 (47 finished feed; 24 feed ingredients)	96% (≥2); 75% (≥5); 13% (≥8)	Regulated (DON, ZEN, FUM, OTA, AF, T-2/HT-2) and non-regulated mycotoxins	Not stated	[64]
Taiwan	Swine feed	820	91.3% (≥2)	(DON, ZEN, FUM, OTA, AF)	ZEN + DON (13.87%); ZEN + FUM + DON (12.47%); AF + ZEN + DON (16.16%); AF + ZEN + FUM + DON (23.54%)	[79]

AFs (AFB_1_, AFB_2_, AFG_1_, AFG_2_); 15-ADON (15-acetyl deoxynivalenol); FUMs (FB_1_, FB_2_, FB_3_); NIV (nivalenol); DAS (diacetoxyscirpenol); FUS-X (fusarenon X); NEO (neosolaniol); AOH (alternariol); AME (alternariol methyether); ROQ-C (roquefortine).

**Table 4 toxins-17-00365-t004:** Impact of multiple mycotoxins on the growth performance of different livestock species.

Livestock	Mycotoxin Combination	Type of Interaction	Effect Compared with Individual Toxin	References
Pigs	DON + ZEN	Synergistic	Greater reduction in weight gain and feed intake than DON or ZEN alone	[120]
Poultry	FB_1_ + DAS/OTA; FB_1_ + AF	Additive	Reduction in body weight, carcass and organ weights	[130,131]
	AF + DON			[113]
	AF +T-2	Synergism		[134]
	AF + DAS			[135]
	OTA + DON	Antagonistic		[134]
Ruminants	AFB_1_ + OTA + ZEN	Additive/Synergistic	Significant drop in DMI and milk yield; individual mycotoxins alone showed minimal effects	[120,125]
Rabbits	AFB_1_ + OTA	Synergistic	40% reduction in weight gain vs. 12% individually; 25% mortality vs. 12.5%	[137]

**Table 5 toxins-17-00365-t005:** Impact of multiple mycotoxins on the immune indices of different livestock species.

Species	Mycotoxin Combo	Interaction Type	Affected System	Effect of Severity vs. Individual Toxin	References
Pig	AFB_1_ + DON	Synergistic	Immune	Stronger immunosuppression; ↓ lymphocyte proliferation, ↓ IgG, IgA, ↓ IL-2, IFN-γ	[138,139]
	ZEN + DON	Synergistic	Reproductive	Severe ovarian dysfunction; ↓ aromatase, ↓ LH receptor expression, aneuploidy	[140]
Broilers	DON + OTA	Synergistic/ Additive	Immune	↓ Macrophage activity, ↓ phagocytosis, ↑ susceptibility to infection	[141,142]
	AFB_1_ + FB_1_	Synergistic/ Additive	Immune and Liver Pathology	Severe liver damage, ↓ NDV antibody titers, complex immune responses	[122,143,144,147]
	OTA + AFB_1_	Synergistic		Histopathological changes and apoptosis in the kidney and liver	[110]
Dairy goats	AFB_1_ + OTA + ZEN (or FB_1_)	Synergistic	Immune and Antioxidant	Elevated liver enzymes (ALT, ALP), ↑ MDA, ↓ SOD, ↓ GSH-Px	[122]
Dairy cows and calves	AFB_1_ + DON + ZEN	Additive/Synergistic	Immune	↑ Somatic cell count, impaired immune function	[149]
	DON + ZEN	Less than additive	Immune	↓ WBC count, mild neutrophil suppression	[148]

**Table 6 toxins-17-00365-t006:** Impact of multiple mycotoxins on the production indices of different livestock species.

Species	Mycotoxin Combo	Interaction Type	Major Production Impacts	References
Dairy cattle	AFB_1_ + DON; AFB_1_ + DON + ZEN	Synergistic	↓ Milk yield (14%), ↑ somatic cell count, ↓ milk fat and protein, reduced DMI, altered rumen fermentation	[125,156]
Dairy goats, cows	AFB_1_ + DON + ZEN; AFB_1_ + OTA; AFB_1_ + OTA + ZEN	Synergistic	↓ Milk yield (18%), ↓ milk protein (12%), synergistic effects between toxins	[112,155]
Sheep	AFB_1_ + OTA	synergistic	↓ Fertility rate, fecundity rate, lambing rate, litter size and number, ↑ still birth	[154]
Broilers/pullets	FUM + DON + ZEN; AF + OTA	Synergistic/Additive/less than additive	↓ Average daily gain, ↑ feed conversion ratio (20%), ↓ egg production and hatchability, egg mass and egg weight	[157,158,160,161]
Pigs (sows)	ZEN + DON	Synergistic	↑ Mean weight of uterus, concentration of FSH	[162]
Pigs (grower–finishers)	DON + ZEN + FB_1_	Synergistic	↑ Backfat thickness, ↓ loin eye area, darker meat colour, ↓ water holding capacity	[104]

## Data Availability

No new data were created or analyzed in this study. Data sharing is not applicable to this article.

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
