# Peer review of "Multiple Mycotoxin Contamination in Livestock Feed: Implications for Animal Health, Productivity, and Food Safety"

_toxins, 2025, doi:10.3390/toxins17080365_

Round 1
Reviewer 1 Report
Comments and Suggestions for Authors
The Review [toxins-3644266] entitled (Multiple-Mycotoxin Contamination in Livestock Feed: Implications for Animal Health, Productivity and Food Safety) focused on the recent findings on multi-mycotoxin contamination in livestock feed, with their physiological effects, mechanisms of action, and implications for regulatory frameworks
The review is presented in a good and comprehensive manner for all aspects related to the title.
- All tables and figure are adequate and well designed
- References are adequate to the study
- In general, the review is well organized and introduced with good writing
Here, my comments:
- The manuscript did not match with the style of MDPI in citation, it should be as [1, 2,…] NOT author names and year
- AS citation, References: Not as MDPI style. They should be written as MDPI style
- In Table 1: Three references are missing (s,t,i). Delete the footnotes and add citations as numbers within the table
- In Table 2, 3, 4, 5 and 6: add citations as numbers within the table
- Lines 119-126: Add references
- Abbreviate the genera names after the first mention of the species as line 172, the species was mentioned in line 73. Check all the text
- Add references for lines 263-267 and 275-278 and 323-335
Author Response
Comment 1: The manuscript did not match with the style of MDPI in citation, it should be as [1, 2,…] NOT author names and year; AS citation, References: Not as MDPI style. They should be written as MDPI style
Response 1: The entire manuscript has been formatted to the style of MDPI
Comment 2: In Table 1: Three references are missing (s,t,i). Delete the footnotes and add citations as numbers within the table. In Table 2, 3, 4, 5 and 6: add citations as numbers within the table
Response 2: All the missing references have been included and formatted to the style of MDPI. The citations are now within the table
Comment 3:
- Lines 119-126: Add references
- Abbreviate the genera names after the first mention of the species as line 172, the species was mentioned in line 73. Check all the text
- Add references for lines 263-267 and 275-278 and 323-3
Response 3: References have been added to Lines 119-126; 63-267 and 275-278 and 323-3; Abbreviations have been done to all the species after first mention
Reviewer 2 Report
Comments and Suggestions for Authors
Reviewers comment
The manuscript is well written whereas some minor changes need to be done
Certain references used in this manuscript are very old please replace them with updated once.
Line 64: Aspergillus flavus
this is already mentioned in the introduction section no need to mention the full species name again in the document. Check it once
Line 74: FUMs include FB1, FB2, and FB3, with FB1 being the most toxic (Li et al., 2022). FB1 exhibits hepatotoxic and nephrotoxic properties and is carcinogenic in animals (Abdul and Chuturgoon, 2021). Among livestock, pigs show the highest sensitivity, while poultry and ruminants exhibit relative resistance. In pigs, FB1 exposure can induce pleural effusion, pulmonary oedema, and pancreatic damage
Either use subscript or normal script. It needs to be the same throughout the document.
Line 102: Aspergillus ochraceus and Penicillium verrucosum,
this is already mentioned in the introduction section no need to mention the full species name again in the document. Check it once
Table 1: pig>poultry>ruminant>fisha
Pig needs to be in the capital like the others
Line 643: A holistic, integrated approach that includes improved agronomic practices, novel detoxification strategies, and updated regulatory standards is essential for effectively mitigating the complex challenges posed by mycotoxin co-contamination.
Check the grammar of this sentence
Section 3.2
The section doesn’t have any reference for support
Section 4.2
Some sentences (e.g., those describing multi-toxin effects) are quite long; breaking them up could enhance readability.”
Section 4.3
Terms such as “significant”, “highly sensitive”, and “severely affects” should be accompanied by quantitative data or statistical significance where available to strengthen scientific validity
The manuscript does a good job highlighting synergistic effects of multi-mycotoxin exposure, but the mechanistic explanations for these interactions could be expanded (e.g., immunosuppression, gut barrier damage).
Consider adding a brief comparative summary or concluding statement that highlights which species or production indices are most vulnerable based on current literature
The sentence referring to “Table 6” should briefly state what parameters are summarized (e.g., "e.g., milk yield, egg production, growth rate") and ensure that Table 6 is accurately formatted and labeled for ease of cross-reference
Consider rephrasing complex sentences for better comprehension. For example, break long sentences like in lines 582–584 into two parts for clarity
Author Response
Comment 1: Certain references used in this manuscript are very old please replace them with updated once
Response 1: Where necessary, most of the old references have ben replaced with recent updated once
Comment 2: Line 64: Aspergillus flavus this is already mentioned in the introduction section no need to mention the full species name again in the document. Check it once.
Response 2: Line 62, this has been abbreviated and corrected
Comment 3: Line 74: FUMs include FB1, FB2, and FB3, with FB1 being the most toxic (Li et al., 2022). FB1 exhibits hepatotoxic and nephrotoxic properties and is carcinogenic in animals (Abdul and Chuturgoon, 2021). Among livestock, pigs show the highest sensitivity, while poultry and ruminants exhibit relative resistance. In pigs, FB1 exposure can induce pleural effusion, pulmonary oedema, and pancreatic damage. Either use subscript or normal script. It needs to be the same throughout the document.
Response 3: This has been corrected as subscript throughout the manuscript.
Comment 4: Table 1: pig>poultry>ruminant>fish. Pig needs to be in the capital like the others
Response 4: Corrected
Comment 5: Line 643: A holistic, integrated approach that includes improved agronomic practices, novel detoxification strategies, and updated regulatory standards is essential for effectively mitigating the complex challenges posed by mycotoxin co-contamination. Check the grammar of this sentence.
Response 5: This has been re-worded (Lines 703-706).
Comment 6: Section 3.2. The section doesn’t have any reference for support
Response: Relevant and updated references now added
Comment 7: Section 4.2. Some sentences (e.g., those describing multi-toxin effects) are quite long; breaking them up could enhance readability.”
Response 7: This has been broken down into two paragraphs
Comment 8: Section 4.3 Terms such as “significant”, “highly sensitive”, and “severely affects” should be accompanied by quantitative data or statistical significance where available to strengthen scientific validity
Response 8: I have tried to add quantitative data as much as I can, including stating the inclusion levels of the mycotoxins.
Comment 9: The manuscript does a good job highlighting synergistic effects of multi-mycotoxin exposure, but the mechanistic explanations for these interactions could be expanded (e.g., immunosuppression, gut barrier damage).
Response 9: Actually, each of the mycotoxins has its own mechanism of action, especially its negative impacts on animals and can be explained (briefly stated in the introduction part of the manuscript). However, the mechanism of interactions of two or more mycotoxins combined is difficult to explain and would require more extensive research. More difficult is the antagonism resulting two mycotoxins, where the outcome is less detrimental as compared to their damaging individual effects.
Comment 10: Consider adding a brief comparative summary or concluding statement that highlights which species or production indices are most vulnerable based on current literature.
Response: While we acknowledge the importance of summarizing species-specific vulnerabilities, a definitive comparative conclusion is challenging due to several interrelated factors. First, susceptibility to specific mycotoxins varies by species—for instance, pigs are particularly sensitive to zearalenone, whereas poultry are more affected by aflatoxins and trichothecenes. Second, the physiological status of the animal (e.g., age, reproductive stage, immune competence) significantly influences the toxicological response. Third, in many cases, observed health and productivity outcomes result from the combined effects of multiple co-occurring mycotoxins. These multimycotoxin interactions—often synergistic or additive—are highly variable across feed types and geographic regions and are not always fully quantified in published studies. Consequently, while trends exist, drawing broad generalizations across species and production parameters may oversimplify the complexity inherent in multimycotoxin exposure.
Comment 10: The sentence referring to “Table 6” should briefly state what parameters are summarized (e.g., "e.g., milk yield, egg production, growth rate") and ensure that Table 6 is accurately formatted and labeled for ease of cross-reference.
Response 10: This has been addressed
Comment 11: Consider rephrasing complex sentences for better comprehension. For example, break long sentences like in lines 582–584 into two parts for clarity
Response 11: This has been broken down into two paragraphs.
Reviewer 3 Report
Comments and Suggestions for Authors
Dear authors,
the manuscript is well written and from high importance not only for scientist, but also for the regulatory authorities for a better and more accurate risk assessment.
However, I have some minor suggestions and comments:
Line 72: Instead of disease vulnerability use disease susceptibility
Line 80: ZEA can cause some genotoxic effects (e.g., DNA damage in vitro), but this is less central than its endocrine-disrupting effects. So maybe you could change the sentence a bit to:
It disrupts gonadal and endocrine functions in both humans and animals, with additional evidence suggesting genetic, hepatic, and immune toxicity (Balló et al., 2023).
Line 94: Anorexia nervosa is a psychiatric eating disorder in humans, not a symptom caused by mycotoxins. Better wording just anorexia.
Table 1: Reference (i) is missing below the table
In Table 2: Aflatoxins in USA: valid for all AFBs not only for AFB1 as for the EU – maybe you can point that out
Different short-cuts for zearalenone are used. At the beginning ZEA, switching later to ZEN. So please unify (preferably to ZEN).
Graphical presentation of the tables is not optimal, some words have slipped or are very close together. Maybe it is better to use the landscape portrait for the tables.
In the section 3.1. the title AFB1 toxicity – small letter t (the others t is T)
Line 3.4.7. you mention that even low doses led to a synergistic effect, without mentioning the concentration (Qing et al. 2022). This paper for example, is also not mentioned in any of the tables.
For me section 3.2.1 is very general without detailed information and partly redundant.
Line 351: you write emerging mycotoxins can intensify the effect with regulated mycotoxins and then mentioning DON and FB1 – none of these is an emerging mycotoxin.
Line 436: Fusarium – not written in italics
Line 451 and 478: AFB1 – 1 subscript is missing, also later in section 4.2.
Line 476: Fusarium – not written in italics
Table 5: summarizes the effects on livestock species, but you include rabbits here. I would delete this species at least in the table, as rabbits are not classical livestock animals.
References:
Ghareeb, K., Awad, W. A., Zebeli, Q., Böhm, J., 2016. Impact of Fusarium toxins on poultry health and performance. World Poult. Sci. J. 72(3), 559–572.:
I was interested in this publication but was not able to find anything about it. Could you please send me the file?
Rerference Vermin not found – maybe you mean Verma et al?
Alvito, P., Silva, L. J. G., Pereira-da-Silva, L., Loureiro, C., 2010. Assessment of mycotoxins and their risks for human and animal health. Toxins, 2(5), 1225–1239. – not found online
Please re-check all the references once again!
General:
What is missing everywhere are the concentrations used in the combinations, which is really important and add a lot of value to the context. Please add the used concentrations into the tables and also in the text it is important to discuss the relevance.
Author Response
Comment 1: Line 72: Instead of disease vulnerability use disease susceptibility
Response: Line 69: corrected
Comment 2: Line 80: ZEA can cause some genotoxic effects (e.g., DNA damage in vitro), but this is less central than its endocrine-disrupting effects. So maybe you could change the sentence a bit to:
It disrupts gonadal and endocrine functions in both humans and animals, with additional evidence suggesting genetic, hepatic, and immune toxicity (Balló et al., 2023).
Response 2: Corrected. Lines 80-82
Comment 3: Line 94: Anorexia nervosa is a psychiatric eating disorder in humans, not a symptom caused by mycotoxins. Better wording just anorexia.
Response 3: Line 96. the word nervosa removed
Comment 4: Table 1: Reference (i) is missing below the table
Response 4: Missing reference added
Comment 5: In Table 2: Aflatoxins in USA: valid for all AFBs not only for AFB1 as for the EU – maybe you can point that out
Response 5: I did state in the Table 2 that the values quoted are both for total aflatoxins or AFB1.
Comment 6: Different short-cuts for zearalenone are used. At the beginning ZEA, switching later to ZEN. So please unify (preferably to ZEN).
Response 6: I have corrected the short-cut by sticking to ZEN throughout the manuscript.
Comment 7: Graphical presentation of the tables is not optimal, some words have slipped or are very close together. Maybe it is better to use the landscape portrait for the tables.
Response 7: I think it is the way the journal formatted the tables. I am sure the tables would come out better in the final version.
Comment 8: In the section 3.1. the title AFB1 toxicity – small letter t (the others t is T)
Response 8: Changed
Comment 9: Line 3.4.7. you mention that even low doses led to a synergistic effect, without mentioning the concentration (Qing et al. 2022). This paper for example, is also not mentioned in any of the tables.
Response 9: All the mycotoxins' concentration have been included and the paper (Qing et al. 2022) now mentioned in the Table.
Comment 10: Line 351: you write emerging mycotoxins can intensify the effect with regulated mycotoxins and then mentioning DON and FB1 – none of these is an emerging mycotoxin
Response 10: Line 379-380. This has been corrected
Comment 11: Line 436: Fusarium – not written in italics; Line 451 and 478: AFB1 – 1 subscript is missing, also later in section 4.2. Line 476: Fusarium – not written in italics
Response 11: All the word Fusarium have been italicized. AFB1 written in subscript.
Comment 12: Table 5: summarizes the effects on livestock species, but you include rabbits here. I would delete this species at least in the table, as rabbits are not classical livestock animals.
Response 12: Rabbit has been removed from the table
Comment 13: References:
Response 13: All the missing references included or replaced
Comment 14: General:
What is missing everywhere are the concentrations used in the combinations, which is really important and add a lot of value to the context. Please add the used concentrations into the tables, and also in the text it is important to discuss the relevance.
Response 15: All the mycotoxin concentrations have now been included.
Reviewer 4 Report
Comments and Suggestions for Authors
This article provides a comprehensive assessment of mycotoxin contamination in food and feed, highlighting the growing concern about the emergence of multi-mycotoxins due to environmental factors, fungal co-infection and global trade. It effectively highlights the main mycotoxins affecting feed, with a strong focus on their toxicological impact on animals and the subsequent risks posed to human health through animal products. One of the key contributions of the study is the focus on polytoxicity, highlighting how mycotoxins rarely occur in isolation and often exhibit additive, synergistic and sometimes antagonistic toxic effects. This is particularly significant as current regulatory frameworks largely address individual toxins, overlooking the complexity of interactions with multiple mycotoxins. The analysis highlights the need for revised risk assessment models that take into account cumulative toxicity, rather than relying solely on thresholds for isolated toxins.
- Subchapter 2.1. In the case of Aflatoxins, the main crops in which they occur are also specified. It must also be specified for the other mycotoxins which are the crops in which they frequently occur.
- Subchapter 2.3.1 Considering that a large part of mycotoxins appear during storage and that most raw materials for feed processing are stored for a long time, I believe that it should be presented in more detail how storage favors the appearance of mycotoxins.
- Line 221-224 Considering that regulated mycotoxins are specified, they should be presented under table 3
- The additive and synergistic effects of mycotoxins that occur concurrently are presented, but perhaps some methods for quantifying these complex interactions should also be mentioned.
- Although continuous monitoring of mycotoxin infestation is suggested, implementing accurate and cost-effective detection methods in global feed supply chains can create logistical difficulties. It would be useful to present in the article more accessible analytical techniques for mycotoxin determination that would improve practical application
Author Response
Comment 1: Subchapter 2.1. In the case of Aflatoxins, the main crops in which they occur are also specified. It must also be specified for the other mycotoxins which are the crops in which they frequently occur
Response 1: The main crops for other mycotoxins explained have been included
Comment 2: Subchapter 2.3.1 Considering that a large part of mycotoxins appear during storage and that most raw materials for feed processing are stored for a long time, I believe that it should be presented in more detail how storage favors the appearance of mycotoxins.
Response 2: A detailed explanation of how mycotoxins appear during storage and in most raw materials for feed processing are now included in the manuscript. Lines 157-196
Comments 3: Line 221-224 Considering that regulated mycotoxins are specified, they should be presented under table 3
Response 3: We have now specified the 'regulated mycotoxins' in Table 3. The word 'regulated mycotoxins' has been removed and replaced with the detailed mycotoxins.
Comments 4: The additive and synergistic effects of mycotoxins that occur concurrently are presented, but perhaps some methods for quantifying these complex interactions should also be mentioned.
Response 4: The method for quantifying mycotoxins, including other analytical methods, is currently being developed in another review manuscript. Adding them to this manuscript might increase the word count beyond the acceptable limit of the journal.
Comment 5: Although continuous monitoring of mycotoxin infestation is suggested, implementing accurate and cost-effective detection methods in global feed supply chains can create logistical difficulties. It would be useful to present in the article more accessible analytical techniques for mycotoxin determination that would improve practical application.
Response 5: This is true and it is part of what would be included in the new review manucript on 'Recent analytical techniques for mycotoxins detection in finished feed and feed samples' .
Round 2
Reviewer 3 Report
Comments and Suggestions for Authors
Dear authors,
thanks for including my suggestions - I think now it is a really valuable manuscript.